

**Distribution Patterns and Community Assembly Processes of Eukaryotic Microorganisms in**

**Tibetan Plateau Proglacial Lakes at Different Emergence Stages**

**Authors**

Jinlong Cui[1,2,3], Qianggong Zhang[2,4*], Qing Yang[3], Fuyuan Mai[1], Shengnan Li[2,4], Mingyue Li[2,4], Jie

Wang[2,4], Xuejun Sun[5], Yindong Tong[1,3**]

**Affiliations**

[1]School of Ecology and Environment, Tibet University, Lhasa 850000, China

[2]State Key Laboratory of Tibetan Plateau Earth System, Environment and Resources (TPESER),

Institute of Tibetan Plateau Research, Chinese Academy of Sciences, Beijing 100101, China

[3]School of Environmental Science and Engineering, Tianjin University, Tianjin 300350, China

[4]University of Chinese Academy of Sciences, Beijing 100049, China

[5]School of Environment and Resources, Shanxi University, Taiyuan 030000, China

**\*Corresponding authors**

\*Qianggong Zhang, email at qianggong.zhang@itpcas.ac.cn

\*\*Yindong Tong, email at yindongtong@tju.edu.cn

**Abstract**

Proglacial lakes are rapidly expanding due to climate change and glacier retreat. Eukaryotic

microorganisms play a crucial role in the biogeochemical cycles of these lakes. However, there is

limited understanding of the formation processes of eukaryotic microbial communities and their

responses to material cycling in proglacial lakes, which are land reservoirs and new habitats for

biological evolution of glacier meltwater, particularly in proglacial lakes at different developmental



stages. This study investigates the distribution patterns and community assembly process of
eukaryotic microbes in high altitude proglacial lakes, formed during different periods (i.e., 1990s,
2000s and post-2010), located on the central Tibetan Plateau. Using 18S rDNA gene amplification
sequencing, in conjunction with neutral community model and a null model, we analyze the spatial
dynamics and assembly processes of eukaryotic microbial communities. Our results reveal
significant spatial heterogeneity community structure. Characterized by a pronounced geographical
distance-decay pattern that intensifies with the age of the proglacial lake, indicating stronger
symbiotic relationships and biological nesting. For proglacial lakes formed at different times,
ecological shifts account for approximately 80% of the observed community variations. Water
temperature was the primary environmental factors influencing the formation of eukaryotic
microbial communities. This study provides valuable data on the distribution patterns and assembly
processes of eukaryotic microbial communities in emerging proglacial lakes, enhancing our
understanding in the trajectories of eukaryotic microbial communities' formation in high altitude
glacier lakes in the context of climate change, and offering insights into the mechanisms that sustain
eukaryotic microbial diversity in extreme environments.
**Keywords**
Proglacial lakes; eukaryotic microorganisms; distribution pattern; community assembly
process; glacier retreat; Tibetan Plateau
**1. Introduction**
Understanding and adapting to global climate change is one of the greatest challenges facing
humanity in the 21st century (Zhang et al., 2024). The world is currently experiencing accelerated
climate warming, with high-latitude and high-altitude regions being particularly sensitive (Andresen



et al., 2011; Lowell, 2000). In these regions, glaciers are highly sensitive to climate change, and
their melting has contributed to the expansion of ice-marginal, moraine-dammed, and supraglacial
lakes (Otto et al., 2022; Zhang et al., 2024; Zhi-Guo, 2012). between 1990 and 2018, the global
volume of glacial lakes increased by approximately 48%, reaching 156.5 $km^3$(Shugar et al.,
2020).For example, one study on the ice phenology of over 13,300 lakes in the Arctic region
between 2000 and 2013 found that all the regions found a clear trend of earlier melting (Salerno et
al., 2014). In China, there were 18,325 glacial lakes with an area of 1185.9 $km^2$, in 2020, an increase
of about 18% compared to 1990(Yin et al., 2023). Of these, a total of 5,894 glacial lakes were located
on the Tibetan Plateau, with an area of 784.8 ± 41.2 $km^2$ and a volume of 20.1 ± 17.1 $km^3$.
Additionally, 869 of these lakes were classified as proglacial lakes (207.3 ± 8.2 $km^2$ and 10.4 ± 9.4
$km^3$)(Zhang et al., 2023). Global glaciers cover about 705,253 $km^2$, with those on the Qinghai-Tibet
Plateau accounting for 49,873 $km^2$, or about 7.1%(Farinotti et al., 2019; Yan et al., 2020). Glaciers
on the Tibetan Plateau lost 22% of their coverage between 1977 and 2010, and the annual shrinkage
rate accelerated in recent decades compared with the previous time period of 1977–2001(Xu et al.,
2013). As global and regional glaciers have melted at an accelerated pace, the global proglacial lake
area has increased by about 11%(Zhang et al., 2024), while the area on the Qinghai-Tibet Plateau
has expanded by 18.4%(Zhang et al., 2017), expanding at nearly twice the global rate. Over the past
few decades, both the area and number of these lakes have significantly increased(Yin et al., 2023).
For instance, research shows that since the 1990s, the area of proglacial lakes on the Tibetan Plateau
has expanded by approximately 10% to 30%, the number of glacial lakes has risen from 15,492 in
1990 to 18,235 in 2020(Yang et al., 2018; Yin et al., 2023). In addition, glacial meltwater impacts
the temperature, turbidity, transparency, and thermal stratification of lake water, as well as the levels





of dissolved organic matter, pollutants, ions, and heavy metals, thereby influencing the composition
and diversity of glacial lake organisms(Wang et al., 2019). This expansion of proglacial lakes is not
only altering regional hydrological dynamics but also raising concerns about potential impacts on
ecosystems. Therefore, studying these proglacial lakes was essential for understanding their role in
extreme environmental contexts, particularly their potential impacts on microbial community
dynamics and shifts in ecological niches(Shu and Huang, 2022; Teittinen et al., 2023).

The formation of proglacial lakes introduced greater complexity and diversity into the material

cycling processes within glacial environments, with eukaryotic microorganisms playing a vital role.
However, much of the current research has largely concentrated on the physical growth of these
lakes and the associated risk of outburst floods(Nie et al., 2021; Worth and Jess, 2009), with less
emphasis on the role of proglacial lakes in the emergence and community assembly of eukaryotic
microorganisms. Eukaryotic microbes, as key components of primary producers, play a significant
role in the biogeochemical cycles of key elements such as carbon, nitrogen, and sulfur(Filker et al.,
2016b; Zhao et al., 2022). For most of these organisms, solar radiation was the primary energy
source, harnessed through photosynthesis(Cui et al., 2023). However, the high turbidity of glacial
lakes reduced the absorption of light and radiation, forcing these microorganisms to rely on chemical
substances to fuel their biological processes. Heterotrophic activity in lake microorganisms was
often sustained by low organic carbon inputs from glacial meltwater or by organic matter stored in
marine sediments beneath the ice sheet(Hood et al., 2015; Wadham et al., 2012). Chemolithotrophy
has also been identified as a significant energy pathway for these organisms(Vick-Majors et al.,
2016). Mixotrophic symbiosis, a common interspecies metabolic relationship in extreme
environments, enables microorganisms to collaborate and catalyze a range of biogeochemical





reactions, allowing them to adapt to harsh conditions(Anantharaman et al., 2016; Ino et al., 2018).
Furthermore, the interactions between prokaryotes and eukaryotes form a tightly integrated
metabolic network, effectively coupling the carbon, nitrogen, and sulfur cycles in glacial lake
ecosystems(Vick-Majors et al., 2014). The eukaryotic microbes communities and its structural and
diversity characteristics can undergo significant changes across proglacial lakes formed during
different time periods (Bagshaw et al., 2006). In the limited number of studies on microorganisms
in glacial lakes, the most have primarily focused on the spatial and temporal variations in microbial
composition (Hernández-Avilés et al., 2018; Weckström et al., 2018). However, there has been
relatively little investigation into the distribution patterns of microbial communities and their
assembly processes across glacial lakes formed during different time periods.
Eukaryotic microorganisms are essential to the development of glacial lake ecosystems,
playing a crucial role in the cycling of materials and the flow of energy within these environments.
They are key contributors to the biogeochemical processes that sustain these fragile
ecosystems(Stock et al., 2022). Eukaryotic microorganisms actively contribute to production and
degradation of organic matter, influencing the cycling of organic matters in the glacial lakes (Zhou
et al., 2019). In extreme environments like glacial lakes, eukaryotic microorganisms such as algae
and fungi play a crucial role in maintaining ecological balance through processes like photosynthesis,
chemoautotrophy, and organic matter decomposition, serving as a vital source of energy and
nutrients for the upper levels of the food chain(Peay et al., 2016; Rochera et al., 2017). Unlike other
microorganisms, they are capable of thriving under harsh conditions, including low temperatures,
limited light, and high salinity, granting them a distinctive ecological role(Shu and Huang, 2022).
Eukaryotic microorganisms possess complex cellular structures, diverse modes of reproduction,



flexible energy metabolism, robust gene expression regulation, and exhibit high sensitivity to
environmental changes(Filker et al., 2016a; Ortiz-Alvarez et al., 2018). Consequently, eukaryotic
microorganisms are increasingly considered reliable indicators for assessing the environmental
condition of freshwater aquatic ecosystems(Borics et al., 2014; Hering et al., 2018). The continuous
formation glacial lakes in the context of glacier melting provides a unique setting for the succession
of eukaryotic microbial community. The substances carried by glacial runoff serve as a nutrient
source for microbial metabolic activities in glacial lakes, despite their low
concentrations(McCutcheon et al., 2021; Warner et al., 2017). Eukaryotic microorganisms can
usually exhibit stronger growth and metabolic capabilities than prokaryotes in extreme
environments due to their complex structures. In low-temperature glacial lakes, they adapt through
mechanisms like synthesizing low-temperature enzymes and developing specialized membrane
structures, allowing them to sustain growth and diversity(Bock et al., 2018). Newly formed
proglacial lakes are characterized by high turbidity, which may limit photosynthesis of eukaryotic
microorganisms(Slemmons et al., 2013). A study on three lakes along a turbidity gradient found that
diversity and community compositions change significantly when hydrological connectivity to the
glaciers is lost and lakes become clear(Peter and Sommaruga, 2016). Freimann et al., investigated
the spatio-temporal patterns of main bacterial groups in alpine water and founded that several
physic-chemical variables which reflect the local geological characteristics and water source,
influence the structure of the bacterial groups(Freimann et al., 2015). These studies primarily focus
on how microbial community composition and diversity respond to environmental changes.
However, the ecological processes underlying microbial community distribution patterns are still
poorly understood. Studying the assembly process of eukaryotic microbial communities helps us



understand the competition, symbiosis, and interactions between microbial species, providing
deeper insight into how these interactions sustain the community's function and the stability of the
ecosystem. Additionally, it reveals how communities respond to environmental changes (such as
temperature, light, and salinity), enabling better prediction and management of ecosystem dynamics
and evolution under varying environmental conditions. Since biogeographical patterns, which
center on geographic and environmental distances, are a core element of ecology, understanding
community assembly mechanisms is essential (Filker et al., 2016b). In this context, In this context,
identifying key taxa and analyzing their interactions through co-occurrence network topology can
offer valuable insights into the processes shaping microbial communities (Xianrong Li et al., 2022).
Two processes have been proposed to explain microbial community changes: deterministic
processes (i.e., niche theory) and stochastic processes (i.e. neutral model theory)(Stegen et al., 2013).
These two processes have been shown to play key roles in various ecosystems or biological types
(Jiao et al., 2020). The niche theory holds that the formation and dynamic changes of microbial
communities are mainly affected by a series of decisive factors, including both abiotic factors (e.g.,
environmental conditions such as pH, temperature, and oxygen concentration) and biotic factors
(e.g., competition between species, mutualistic symbiosis, and predation). These factors work
together to determine the distribution, reproduction, and mutual relationship of different microbial
species in the ecosystem(Dumbrell et al., 2010). For example, Dumbrell et al. suggested that AM
fungal communities are strongly influenced by environmental factors and that they responded
predictably and deterministically to changes in pH(Dumbrell et al., 2010). In contrast, neutral theory
holds that changes in species diversity and community structure are driven primarily by random
processes (e.g., the birth, death, and migration of species) rather than by natural selection or

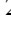



ecological adaptation. A study found that stochastic processes played a key role in shaping the
assembly of microeukaryotic communities in a subtropical river during both the wet and dry seasons
(Chen et al., 2019).
Proglacial lakes on the Tibetan Plateau are typically located at high-altitude regions (e.g.,
5000–5500 m a.s.l.)(Zhang et al., 2015), and are characterized by extremely cold conditions, with
an annual average temperature of -10 °C(Zhang et al., 2015). With minimal disturbance from human
activities, these proglacial lakes better reflect the natural state of microbial community changes
(Yang et al., 2023b). During the summer melting periods of glaciers, glaciers transport nutrients and
microorganisms into newly formed proglacial lakes (Fegel et al., 2019), influence the microbial
community composition of these lakes. Over the past decade, accelerated glacier melting and retreat,
numerous proglacial lakes are forming and are becoming an integral part of aquatic
ecosystems(Burpee and Saros, 2020; Scapozza et al., 2019). This highlights the need to comprehend
changes in the water environment of glacial lake and process of constructing aquatic community, as
these are crucial for understanding the biogeographic patterns, biogeochemical cycles, and
ecosystem functional changes in the cryosphere region under the background of climate changes.
Notably,the 18S rDNA gene amplicon sequencing technology, through the amplification and
sequencing of the 18S rRNA gene, provides us with diverse information about eukaryotic microbial
communities, including species diversity, community structure, environmental adaptability,
functional analysis, and ecological change monitoring(Cui et al., 2023). This technology can
identify the eukaryotic microorganisms in environmental samples, assess the species composition
and diversity of the community, and reveal the interactions among microorganisms within the
community. Additionally, it can uncover how microorganisms adapt to specific environmental





conditions (such as temperature, salinity, light, etc.), and, by combining other genomic data, infer
the ecological functions of species(Kumar et al., 2021; Muhammad and *, 2021). To explore these
dynamics, we collected surface and bottom water samples from three proglacial lakes formed during
different historical periods (i.e., 1980s, 1990s, and 2010s) on the Tibetan Plateau during the ice-free
periods of May and August in 2021 and 2022. The selection of these lakes was based on their
formation times, which collectively span the period when glacial lakes have emerged most
frequently over the past few decades. This enables a continuous and high-resolution understanding
of the microbial ecological changes within these lakes. Diversity characteristics, biogeographical
patterns, community symbiotic networks, and community assembly processes of eukaryotic
microbial communities in these samples were analyzed using 18S rDNA gene amplicon sequencing
technologies to address the following questions: (1) Do eukaryotic microbial communities vary
among proglacial lakes that formed during different time periods? (2) What is the dominant
process — deterministic or stochastic — shaping the assembly of the eukaryotic microbial
communities? Our study aims to elucidate the distribution patterns and assembly processes of
eukaryotic microbial communities, providing new insights into the diversity characteristics of
eukaryotic microbes in emerging proglacial lakes on the Tibetan Plateau and contributing to a better
understanding of these characteristics globally.
**2. Materials and methods**
**2.1 An overview of the study area**

Mount Nyenchen Tanglha is located at the southern part of the Tibetan Plateau home to 7,080

glaciers spanning 10,701 km$^2$(Ji et al., 2014; Tao et al., 2021). Various studies have reported rapid
glacial retreat and the formation of glacial lakes in this region(Wang et al., 2012). Kuoqionggangri





Glacier (KQGRG, latitude 29º5', longitude 90º12', altitudes of 4,800-6,200 m) is one of the typical
glaciers in the Mount Nyenchen Tanglha region. During the summer, this region is primarily
influenced by the Indian Ocean monsoon while in winter, it is dominated by the westerly winds(Cui
et al., 2023). The annual average rainfall in this region is 457−581 mm, and the annual average
temperature is -6.4 º C, with 16.52 º C in summer and -20.12 º C in winter(Cui et al., 2023). This
glacier basin has remained largely unaffected by significant human activities, making it an ideal
area for evaluating the impact of climate change on aquatic ecosystems (Cui et al., 2023). Since the
1990s, proglacial lakes have expanded rapidly in the KQGRG basin(Xu et al., 2023). In this study,
we focused on three accessible proglacial lakes that originated from the KQGRG (Fig. S1). These
lakes were found to have formed at different time periods, specifically in the 1980s, 1990s, and
2010s, as identified through analyses of Google Earth images (Fig. S1)(Sun et al., 2022). Lake 1
(latitude: 29°52'2"; longitude: 90°11'48") is identified as a newly formed lake that likely originated
around the 2010s, referred to as Newly-emerging Lake (NL). It has an area of 3,500 m² and a depth
of 3 m. It is directly fed by the glacial runoff. Lake 2 (latitude: 29°51'57"; longitude: 90°12'8") is
directly supplied by glacial runoff. It originated around 1990 and is referred to as Middle-period
Lake (ML). It spans an area of 5,000 m² with a depth of 6 m. Lake 3 (latitude: 29°53'11"; longitude:
90°11'29") formed before 1980, referred to as Early-emerging Lake (EL). It has an area of 6,500
m² and a depth of 2 m (Fig. S1). Significant differences in temperature, dissolved oxygen, light, and
nutrient availability across lake depths can drive changes in microbial community composition,
metabolic pathways, and ecological functions(J et al., 2003). For example, shallow water layers,
with higher temperatures and ample light, promote the growth of photosynthetic microorganisms,
while deeper layers, characterized by lower temperatures and reduced oxygen, may favor




chemoautotrophic organisms or those relying on anaerobic metabolism(Rose et al., 2009).
Additionally, nutrients from glacier meltwater are typically concentrated in surface waters, while
deeper layers often have fewer nutrients, potentially influencing the structure of microbial
communities at varying depths(Sommaruga, 2015). Therefore, studying lakes at different depths
offers valuable insights into how water depth impacts microbial ecology and ecosystem functioning.
**2.2 Field investigation and sample collection**
We collected biological and water samples from the surface and bottom waters of the three
proglacial lakes during the ice-free periods on May 15th and August 15th, in 2022. For each lake,
water from the inlet, outlet, and the center was collected. Water from three sampling sites were
combined to create a sample of lake. Surface water was collected at a depth of 0.2-0.5 m below the
proglacial lake surface (pls), while bottom water was collected at a depth of 0.2 m above proglacial
lake bottom (plb). From each sampling site, 5 liters of water was collected and filtered using PVDF
membrane (Millipore Millex, 47 mm, 0.22 μ m) to enrich microbial samples. The filtered water was
then used to detect indicators such as dissolved nitrogen and phosphorus. Water temperature (WT),
pH, and electrical conductivity (EC) were measured using a portable water quality parameter meter
(HANNA). Salinity (SAL) was determined using a salinity meter (AZ-8373). Turbidity (TUR) was
measured using a turbidity meter (SGZ-1000BS), and dissolved oxygen DO was analyzed using a
portable dissolved oxygen analyzer (JPB-607). In the laboratory, water quality indicators such as
total nitrogen (TN), nitrate nitrogen ($NO^3$-N), nitrite nitrogen ($NO^2$-N), ammonia nitrogen ($NH^{4+}$-
N), total phosphorus (TP), dissolved total phosphorus (TDP), orthophosphate ($PO_4^{3-}$), particulate
phosphorus (PP), and chlorophyll a (Chl-a) were measured. TN was determined using the alkaline
potassium persulfate oxidation method(Cui et al., 2023). $NO^3$-N was determined using phenol



disulfonic acid photometric method and spectrophotometry was used for the $NO^2$-N
determination(Cui et al., 2023). $NH^{4+}$-N was determined out using a hypobromate oxidation
method(Cui et al., 2023). TP, TDP, $PO_4^{3-}$, and others were determined using phosphomolybdate
heteropoly acid spectrophotometry(Cui et al., 2023). PP was calculated by subtracting TDP from TP.
Chl-a was determined by spectrophotometry(Cui et al., 2023). The detection limit for each
experimental method were available in the Table S1.
**2.3 DNA extraction and high-throughput sequencing**
The 18S rDNA V4 hypervariable region was PCR-amplified using the primers 547F (5'-
CCAG-CASCYGCGGTAATTCC-3') and 952R (5'-ACTTTC-GTTCTTGATYRA-3') (Cui et al.,
2023). PCR products were examined by electrophoresis on a 2 % agarose gel and purified using a
GeneJET Gel Recovery Kit following electrophoresis on a 1 × TAE buffer. The purified amplicons
were pooled equimolarly and subjected to paired-end sequencing on Illumina MiSeq PE300
platform or a NovaSeq PE250 platform (Illumina, San Diego, USA) based on the standard protocols
provided by Majorbio Bio-Pharm Technology Co. Ltd. (Shanghai, China).
**2.4 Data processing and analysis**
Raw reads were processed with Trimmomatic(Magoc and Salzberg, 2011) to remove low-
quality reads (quality score < 20), short reads (< 100 bp), reads that had mismatches with the barcode,
and reads with a maximum of two mismatches to the primer. High-quality paired-end reads were
combined using PEAR(Magoc and Salzberg, 2011). Operational taxonomic units (OTUs) were
clustered at a 3 % dissimilarity level using UPARSE(Edgar, 2013). Singleton and doubleton OTUs,
which represent the sequencing errors, were excluded from the subsequent analyses. Taxonomy
assignment for the OTUs was conducted against the SILVA database using the RDP



classifier(version 2.2)(Wang et al., 2007). To standardize the sequencing depth, each sample was
rarefied to 10,763 reads while preserving all the original sequence information for each sample. The
rarefied sequences were used to calculate alpha diversity (Shannon, Pielou, Simpson and Richness)
in Qiime2(Golshanrad et al., 2021). For beta diversity analysis, the Principal Coordinates Analysis
(PCoA) and permutational multivariate analysis of variance (PERMANOVA) were conducted based
on Bray-Curtis dissimilarities. For environmental factor analysis, Mann-Whitney U Test (M-WUT)
and Wilcoxon Signed Rank Test (WSRT) were used to analyze the differences of environmental
factors. Additionally, the effects of environmental factors and spatial structure on species was
examined using Principal Components of Neighborhood in Manteau (PCNM) and Canonical
Correlation Analysis (CCA). These analyses were carried out using the "vegan" package in
R(version 4.3.1)(CHEN et al., 2022).
Pairwise geographic distances between samples were calculated based on latitude and
longitude coordinates. This calculation was performed using "geosphere" library in R(Yang et al.,
2023b). The resulting pairwise distances(environmental and geographical) were then plotted against
the Bray-Curtis dissimilarities of eukaryotic community using the "ggplot2" package in R(Yang et
al., 2023b). The correlation of the regression curves was calculated to assess the relationship
between Bray-Curtis dissimilarity and geographic distance.
We employed the framework of Stegen et al(Stegen et al., 2013; Stegen et al., 2015) integrating
phylogenetic and null model analyses to discern community assembly processes. This framework
requires significant phylogenetic signal, meaning that phylogenetic distances approximating the
ecological niche differences among taxa(Stegen et al., 2013). The phylogenetic signal was evaluated
using the Mantel correlogram comparing the distance matrices of environmental optima and



phylogeny for all OTUs(Garner et al., 2023). To evaluate the ecological processes, β-mean nearest
taxon distance (βMNTD), which quantifies phylogenetic turnover between samples was calculated
in the R package "picante"(Jiao et al., 2020). The standardized β-nearest taxon index (βNTI)
identified heterogeneous selection (βNTI > 2) and homogeneous selection (βNTI < −2) both
representing deterministic processes(Yang et al., 2023b). Values between −2 and 2 indicated
stochastic processes, including homogenizing dispersal, dispersal limitation, and drift. Raup-Crick
metric ($RC_{bray}$) was used to differentiate these processes, with $RC_{bray} > 0.95$ indicating dispersal
limitation, $|RC_{bray}| < 0.95$ indicating drift, and $RC_{bray} < −0.95$ indicating homogeneous dispersal
(Yang et al., 2023b). Variation partitioning analysis (VPA) highlighted environmental and spatial
factors' relative and combined effects on eukaryotic microbial communities. Significant
environmental factors identified through canonical correspondence analysis and spatial factors
generated via principal coordinates of neighbor matrices analysis were used. The permutation test
was used to evaluated pure effects of environmental (E|S) and spatial (S|E) factors. Partial Mantel
test was conducted to verified VPA results(Liu et al., 2020). All analyses were conducted in R.

Co-occurrence networks were constructed using the "igraph", "Hmisc", and "qvalue" packages

in R software. For a more focused analysis, only OTUs with a relative abundance exceeding 0.01%
across all samples and appearing in over 20% of the samples were included. Spearman's pairwise
correlations were calculated between OTUs, and those with a correlation coefficient $|r| > 0.7$ and p
$< 0.01$ were deemed significant after applying the Benjamini-Hochberg correction for multiple
comparisons(Stegen et al., 2013). To examine the structure of the network, various network-level
and node-level topological features were calculated. Network-level features included mean node
degree, clustering coefficient, average path length, modularity, density, diameter, betweenness





centralization, and degree centralization(Milke et al., 2023; Thébault and Fontaine, 2010). Node-
level features included degree, transitivity, betweenness centrality, and closeness centrality. To
investigate the distance-decay relationships of co-occurrence patterns, a subgraph was extracted
from the meta-community network for each sample. The resulting network was visualized using
Gephi (version0.9.2, https://gephi.org/), an open-source graph visualization platform(Yang et al.,
2023b).
**3. Result**
**3.1 The physical and chemical characteristics of proglacial lakes**
In this study, we conducted environmental factor measurments on the samples, analyzed the
differences in water environments of different proglacial lakes, and reported the environmental
characteristics of pls and plb (Table S2 and S3). Based on the Wavelet Singular Spectrum Transform,
significant differences in the levels of WT, EC, Sal, and nutrient levels were observed among NL,
ML and EL (M-WUT,WSRT, $P < 0.05$). Among them, the overall water quality index was lower
in NL compared to EL. Among the three types of proglacial lakes, the TUR of ML was high and had
a wide range, no significant difference in TUR values observed between NL and EL. PP among the
three types of proglacial lakes was comparable (Fig. S2). For environmental factors between pls and
plb, extremely significant differences in DO were observed across different layers of proglacial
lakes (Table S3). Significant difference in Chla content were found among different proglacial lakes,
with lower Chla content and biomass in NL and ML compared to EL (Fig. s4). The Chla content in
plb was higher than that of pls, which is opposite to the differences in DO. Specific environmental
factor parameters can be found in Table S1. Overall, WT, TUR, and nutrient levels in NL were
relatively low, while the nutrient levels were generally high in EL. Compared with the



environmental differences among different proglacial lakes, the water quality differences between
pls and plb were not significant, and the spatial heterogeneity of water quality in the same lake is
relatively small.
**3.2 Diversity characteristics of eukaryotic microorganisms**
The 18S rDNA sequences of eukaryotic microorganisms in all samples ranged from 206 bp to
542 bp, with an average length of 380 bp. After quality control and screening, a total of 10,763 reads
were obtained, and 1647 OTUs were clustered at a 97 % similarity level. The coverage ranged from
95.4% to 99.9%, indicating that sequencing recovered the diversity of most local species. Significant
differences in Shannon, Pielou, and Simpson indices were observed between NL and EL in the
different proglacial lakes, whereas no statistically significant difference in alpha diversity indices
were found between NL and ML. Significant differences in the Pielou and Simpson indices were
found between pls and plb (Fig. 1 a, b, c, d). SIMPER analysis indicated that the main contributors
to the eukaryotic microbial communities in proglacial lakes of KQGR were *Bacilliophyta*,
*Ciliophora*, and *Cryptomycota*. The main contributing species to the differences in eukaryotic
microbial communities across different layers of proglacial lakes were *Chlorophyta*, *Cercozoa*, and
*Nematoda* (Table S4).



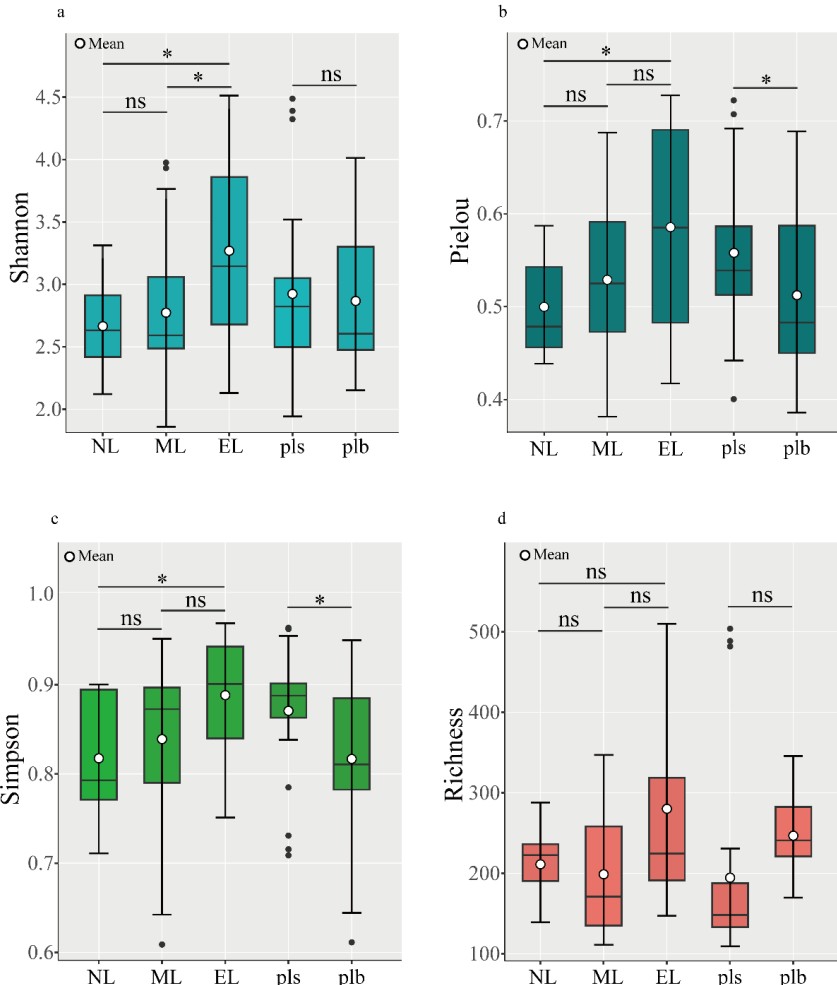

Fig. 1. Diversity indices of eukaryotic microorganisms at different taxonomic levels, respectively

(Note: ***, p < 0.001; **p < 0.01; *p < 0.05, significant difference; ns, no statistically significant).

(NL: Newly-emerging Lake; ML: Middle-period Lake; EL: Early-emerging Lake; pls: proglacial

lake surface; plb: proglacial lake bottom)

The beta diversity analysis of eukaryotic microorganisms. It indicated that the mean β diversity

of eukaryotic microorganisms in NL, ML, and EL was 0.64, 0.75, and 0.72, respectively. The β

diversity of eukaryotic microorganisms in pls and plb was 0.81 and 0.80, respectively. The




proportion of turnover components in NL, ML, and EL ranged from 86 % to 89 %, and turnover
components played a dominant role in β diversity, as do pls and plb (pls turnover accounts for 90 %
and plb turnover accounts for 96 %). Therefore, no obvious nested patterns have formed in different
proglacial lakes or across different levels of proglacial lakes (Fig. 2 a, b, c, d, e).

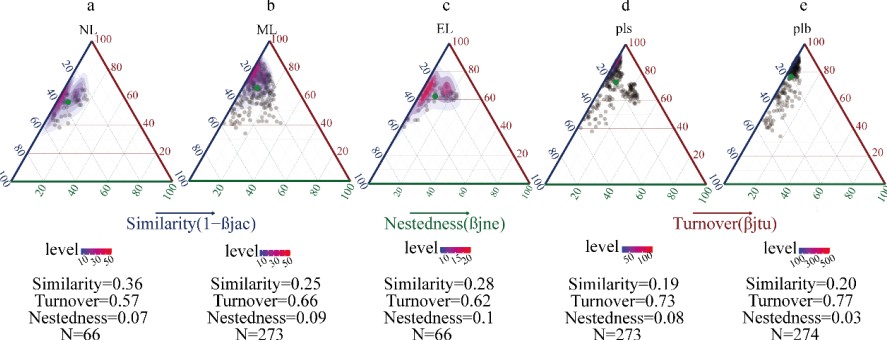


Fig. 2. Decomposition analysis of β diversity components in the eukaryotic microbial community
of the study area. (Representations of similarity, turnover, and nestedness) β the three components
of diversity, n, sample point pairing)
(NL: Newly-emerging Lake; ML: Middle-period Lake; EL: Early-emerging Lake; pls: proglacial
lake surface; plb: proglacial lake bottom)
The PCoA plot based on Bray Curtis distance and PERMANOVA yielded consistent results.
The composition of eukaryotic microbial communities among different proglacial lakes was
comparable (R = 0.639, $P$ = 0.001), and no significant difference was observed in the microbial
communities between pls and plb (R = 0.111, $P$ = 0.005). In PERMANOVA, 'between' represents
inter-group differences, and between-group value was greater than other group level values,
indicating that inter group differences are greater than intra group differences (Fig 3 a, b, c, d). The
Bray-Curtis distance decay of the community indicated a higher level of heterogeneity in
community differences among different proglacial lakes. The correlation between Bray Curtis



heterogeneity and geographic distance was stronger than its correlation with environmental distance.
The changes in eukaryotic microbial communities among different proglacial lakes were greater
than those across different layers of the lake. In samples with closer geographical distances,
eukaryotic microorganisms exhibited a clear distance-decay pattern, where community differences
increased with geographic distance (Fig. 3 e, f, g, h).

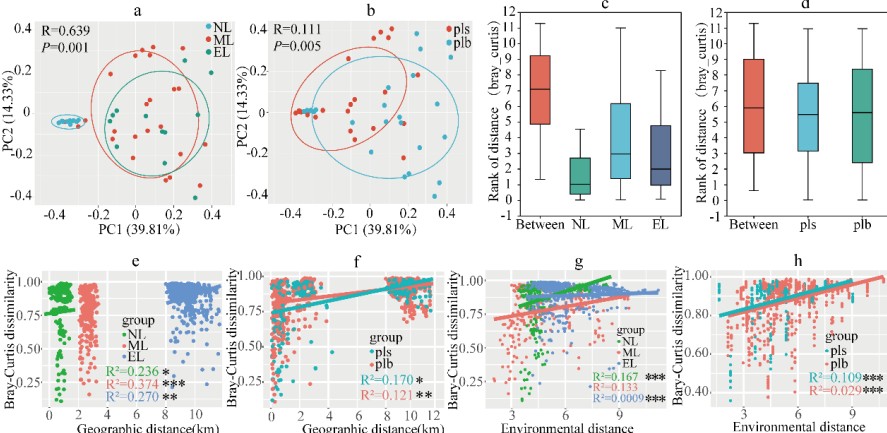


Fig. 3. Panels a and b show phylum-level PCoA for eukaryotic microbial communities. A higher
PERMANOVA R-value indicates greater group differences. P-value < 0.05 signals high reliability.
X-axis represents inter-group distances, Y-axis their magnitude (c and d). Panels e and f show the
relationship between Bray Curtis heterogeneity and proglacial lake evolution time, while g and h
show its link to environmental distance. ***, $p < 0.001$; **, $p < 0.01$; *, $p < 0.05$.
(NL: Newly-emerging Lake; ML: Middle-period Lake; EL: Early-emerging Lake; pls: proglacial
lake surface; plb: proglacial lake bottom)
**3.3 The relationship between eukaryotic microbial communities and environmental factors**

In this study, we used Variance Partitioning Analysis (VPA) to quantitatively evaluate the

contributions of different environmental factors to community composition differences. The results





indicated that TP, TDP, and DO were the main environmental factors influencing community
structure (Fig. 4 a). The environmental factors quantitatively evaluated by VPA explained more than
60 % of the variance for NL, ML, EL, pls, and plb (Fig. 4 b). Additionally, the correlation between
eukaryotic microbial communities and environmental factors was analyzed using the mantel test.
The results showed the strongest correlation between EC, SAL, and NL microbial communities (R >
0.4, $P$ < 0.01), and a strong correlation between $NO_3$-N, WT, EC, DO, and ML microbial
communities (R > 0.4, $P$ < 0.01). No significant strong correlation was observed between the EL
microbial community and environmental factors. The pls microbial community showed the
strongest correlation with $NO_3$-N, WT, EC, and DO (R > 0.4, $P$ < 0.01), while the plb microbial
community has the strongest correlation with SAL (R > 0.4, $P$ < 0.01) (Fig. 4 c and d) (Table S6).
However, no significant correlation was observed between the microbial communities and
environmental factors in EL. Environmental factors showed mainly positive correlations with
eukaryotic microbial communities in NL and ML lakes, but negative correlations in EL lakes.
Nutrients like TN and TP were the primary factors influencing the NL microbial community, while
WT, DO, SAL, and pH were the key factors affecting the ML microbial communities. However, the
EL microbial community was less affected by environmental factors, as indicated by the lack of
significant correlations (Fig. S5).





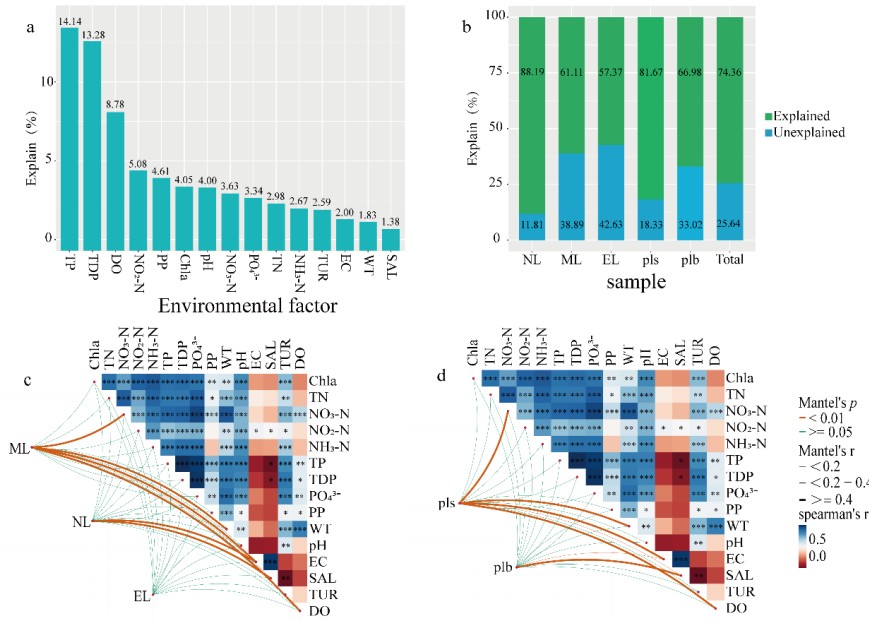


Fig. 4. a illustrates the Variance Partitioning Analysis (VPA) effects of environmental factors on
differences in community structure; b represents the overall explanatory power of environmental
factors at various taxonomic levels in VPA analysis; c and d display the results of the Mantel test
analysis.
(NL: Newly-emerging Lake; ML: Middle-period Lake; EL: Early-emerging Lake; pls: proglacial
lake surface; plb: proglacial lake bottom)
**3.4 Analysis of co-occurrence network of eukaryotic microorganisms**
OTUs with a relative abundance ≥ 0.01 % in each proglacial lake were selected to construct
co-occurrence networks of eukaryotic microbial communities. Topological structural features,
including network nodes and edges at different taxonomic levels, were calculated (Table S5). The
results showed that, in different proglacial lakes, NL had lower average degree and graph density,
indicating a simpler network stracture, EL has higher average degree, graph density level, and lower



average path length, indicating a complex and close interaction relationship between eukaryotic

microorganisms (Fig. 5 a, b, c, d, e). Thus, we conclude that the strength of the symbiotic

relationship between eukaryotic microbial communities in different proglacial lakes was

EL>ML>NL. Similarly, based on the observations of pls and plb, we can conclude that the

interactions among pls species were stronger than those among plb species. The co-occurrence

network results showed that among the proglacial lakes, NL had the fewest coexisting species, while

EL had the most. Among coexisting species, Chlorophyta accounted for the highest proportion

(12.84 %), while Perkinsozoa accounted for the lowest (2.33 %). In different layers, pls contained

more coexisting species, with Chlorophyta accounted for the highest proportion (15.32 %) and

Bacillariophyta accounted for the lowest (1.73 %).

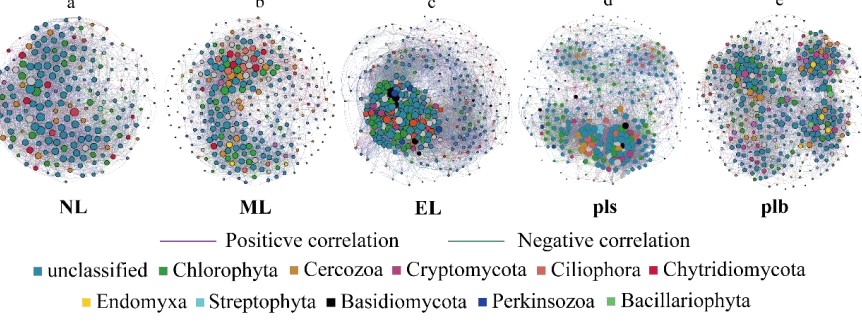

Fig. 5. Analysis of co-occurrence networks at different grouping levels

(NL: Newly-emerging Lake; ML: Middle-period Lake; EL: Early-emerging Lake; pls: proglacial

lake surface; plb: proglacial lake bottom)

**4. Discussion**

**4.1 Spatial Consistency and Variations in Proglacial Lake Water Environment and Eukaryotic**

**Communities**

As previously mentioned, water environments in different proglacial lakes exhibited significant



spatial heterogeneity, but showed negligible differences within different layers of the proglacial
lakes (Fig. S2). Similarly, the diversity of eukaryotic microorganisms in different proglacial lakes
exhibited significant spatial heterogeneity, reflecting similar patterns with the spatial heterogeneity
observed in both cases. The three proglacial lakes followed the same order in terms of diversity,
nesting levels, nutrient concentrations, and water temperature: EL > ML > NL (Fig. 1, Fig. 2, Fig
S3). This indicates that higher nutrient levels result in greater community diversity and stronger
nested patterns. Co-occurrence network analysis revealed that the network edges were
predominantly positively correlated, with the proportion of positive correlations following the same
order: EL > ML > NL (Fig. 5). This suggests that species interactions in the lakes were mainly driven
by synergistic symbiosis, consistent with the findings from β diversity decomposition analysis. In
generally, microbial communities of the same kind often exhibit competitive or antagonistic
interactions. However, in extreme habitats with limited nutrient availability, synergistic effects tend
to dominate, and both community diversity and nutrient levels display similar patterns(Frade et al.,
2020; Karakoç et al., 2018; Liu et al., 2020). This indicates that the interaction dynamics among
different microbial groups within ecological networks were influenced by habitat heterogeneity and
were not static(Peay et al., 2016; Shaffer et al., 2022; Thébault and Fontaine, 2010).
**4.2 Ecological Assembly Process of Eukaryotic Microbial Community in proglacial lakes**
To investigate the mechanisms driving the observed geographic patterns and to clarify the
relative contributions of niche and neutral processes in community assembly, we conducted an
analysis of eukaryotic microorganisms in KQGR proglacial lakes using null and neutral models.
This allowed us to assess the influence of both niche and neutral processes in shaping community
assembly. Significant phylogenetic signals were detected at relatively short distances, indicating





βMNTD was an appropriate distance for measuring phylogenetic turnover(Yang et al., 2023b). The
null model indicates that biological drift was the dominant process, with its proportion exceeding
60 % in NL, ML, and EL (Fig. 6 a, b). Interestingly, biological drift accounted for over 90 % in ML.
The biological drift also accounted for over 70 % in both pls and plb. Although slight variations
were observed among different proglacial lakes, the overall balance of different ecological processes
remained unchanged (Fig. 6a). However, in the surface and bottom layers of proglacial lakes, aside
from biological drift dominating community assembly, significant changes were observed in
homogeneous dispersal and dispersal limitation across different layers. This contrasts with the
relatively stable processes seen between different proglacial lakes (Fig. 6b).

Overall, stochastic processes contribute more to changes in eukaryotic microbial communities

than deterministic processes. To better understand the aggregation process of communities, a neutral
community model was applied. The neutral model (Nm) estimates the product of the
metacommunity (N) size and migration rate (m)(Hubbell, 2001; Leibold and Mcpeek, 2006;
Monchamp et al., 2019). In the proglacial lakes, the Nm value for NL (m = 0.4658) was significantly
higher than that for ML (m = 0.0727) and EL (m = 0.0691), indicating that the dispersal of eukaryotic
microorganisms in NL was higher than in ML and EL. In different layers of the proglacial lakes, the
Nm value for pls (m = 0.0224) was higher than for plb (m = 0.0234), indicating stronger dispersal
of pls microorganisms (Fig 6 a, b). Overall, our results indicate that stochastic processes played a
major role in the assembly of eukaryotic microbial communities, as demonstrated by the null model
analysis. In both different proglacial lakes and layers, community assembly was primarily driven by
stochastic processes, particularly drift. Deterministic processes, such as homogeneous and
heterogeneous selection, played a secondary role, similar to findings in other studies on high-altitude



aquatic microorganisms(Bock et al., 2018; Filker et al., 2016b; Han et al., 2023; Monchamp et al.,

2019).

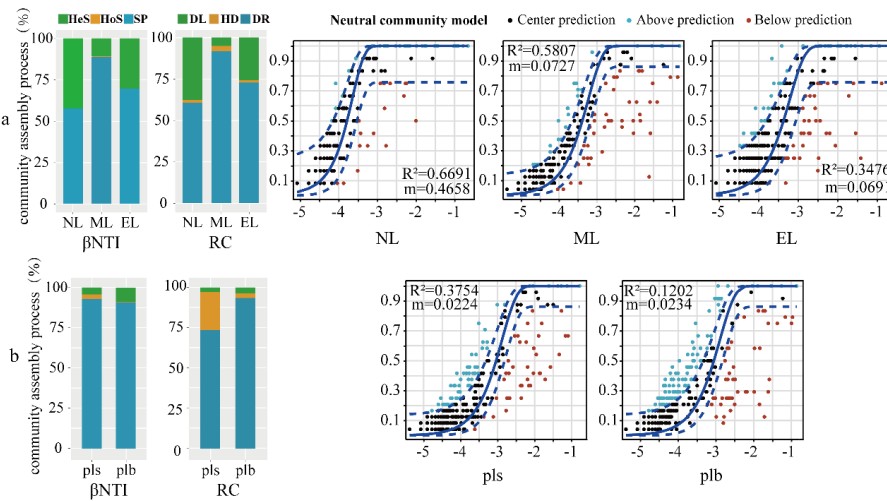

Fig. 6. Panels a and b show ecological process contributions to eukaryotic microbial communities
in proglacial lakes based on null and neutral models. In the Null Model (βNTI: Beta-Nearest Taxon
Index; RC: Raup Crick), βNTI > 2 indicates heterogeneous selection (HeS), βNTI < -2 indicates
homogeneous selection (HoS), and -2 < βNTI < 2 reflects stochastic processes (SP). For SP, RC >
0.95 indicates diffusion limitation (DL), and RC < -0.95 indicates homogeneous diffusion (HD).
For deterministic processes, |RC| ≤ 0.95 represents drift (DR). In the neutral model, random
changes in community structure are shown (the horizontal axis represents average species
abundance, and the vertical axis predicts occurrence frequency), with $R^2$ indicating the model fit,
where higher values indicate better fit to the neutral model. m represents the migration rate; smaller
m values indicate more restricted species diffusion.
(NL: Newly-emerging Lake; ML: Middle-period Lake; EL: Early-emerging Lake; pls: proglacial
lake surface; plb: proglacial lake bottom)



To further identify the intrinsic factors driving community assembly, we employed Mantel tests

(Table S7) to validate the results of the null model. The results indicate that spatial factors (S | E)
have a greater influence than environmental factors (E | S) in different proglacial lakes (Fig. 7a, b).
In different layers of proglacial lakes, Mantel tests show that spatial factors have a significantly
greater influence, consistent with the null model analysis but contrary to the VPA results, which
indicated that environmental factors had a stronger impact. For samples from different layers, the
stability between methods was relatively low, suggesting that spatial factors play a less important
role in these cases. At both grouping levels, more than 70 % of the community variation remained
unexplained, indicating a complex process of community assembly. CCA showed that spatial and
environmental factors significantly affected eukaryotic community assembly in different proglacial
lakes and in different layers of glacial lakes. The CCA analysis revealed that the significant factors
were three spatial factors (PCNM 1-3) and three environmental factors, including WT, water depth,
and SAL (Supplementary Table S7). In different layers of the proglacial lakes, the significant factors
included two spatial factors (PCNM 2 and 3) and two environmental factors (WT and SAL) (Fig.
7c, d).



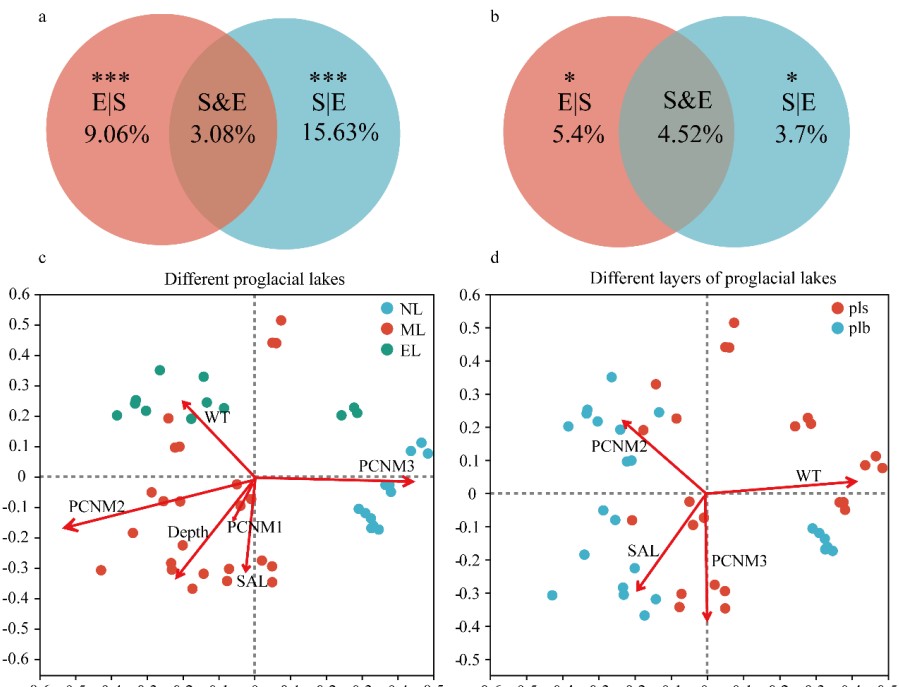

Fig. 7. Using variance partitioning analysis (VPA) and canonical correlation analysis (CCA) to

examine the influence of spatial and environmental factors. VPA quantifies the contributions of these

factors to community variation in different proglacial lakes (a) and in surface and bottom layers (b).

PCNM represents a geographic factor from the principal coordinates of neighboring matrices, and

depth refers to each lake's depth, with dot colors indicating samples from different proglacial lakes.

(NL: Newly-emerging Lake; ML: Middle-period Lake; EL: Early-emerging Lake; pls: proglacial

lake surface; plb: proglacial lake bottom)

**4.3 Spatiotemporal dynamics of the assembly process of eukaryotic microbial communities in**

**proglacial lakes**

We confirmed that spatial factors play a dominate role in shaping the ecological processes of

eukaryotic microbial communities in proglacial lakes. To clarify the potential spatial factors

influencing the community assembly process, we performed regression analyses between RC-bray





and proglacial lake evolution process, along with environmental factors. The correlation between
RC-bray in different glacial lakes and different layers of glacial lakes and the evolution time of
proglacial lakes is stronger. In different proglacial lakes, the correlation was ranked as ML>NL>EL;
in different layers of the proglacial lakes, the correlation was ranked as plb>pls (Fig. 8). In this study,
the three proglacial lakes were situated in distinct geographic locations and had varying
developmental times. NL, which has the shortest development time, was directly connected to the
glacier, allowing it to receive glacier meltwater supply, which enhanced nutrient levels and
promoted microbial drift(Stock et al., 2022; Wang et al., 2019; Yang et al., 2023b). The recently
formed ML was not connected to the glacier but were supplied by glacier meltwater over short
distance and influenced by surface runoff input, which intensified the impact of ecological drift(Shu
and Huang, 2022). The earliest developed EL, surrounded by fully retreated glaciers, was primarily
influenced by rainfall, leading to a weakening of ecological drift(Cui et al., 2023). It can be seen
that the longer the glacial lake evolution time is, the eukaryotic microbial community construction
is affected by the random process, which is first larger and then smaller. Generally, homogeneous
diffusion referred to the uniform distribution of species in space within an environment without
significant environmental or biological heterogeneity, typically driven by migration or
diffusion(Monchamp et al., 2019). As shown in the previous Fig. 1, the surface layer of the
proglacial lake (pls) exhibited greater uniformity compared to the bottom layer (plb), with species
distribution on the surface being more even. This promoted greater mean diffusion, leading to a
higher proportion of homogeneous diffusion on the surface and a lower proportion of biological
drift. Furthermore, in the surface layer of proglacial lakes, the influence of external inputs is stronger,
resulting in a higher diffusion rate compared to the bottom layer(Cauvy-Fraunié and Dangles, 2019;



Nie et al., 2021; Worth and Jess, 2009). At the same time, owing to the substantial input and
disturbance from glacier meltwater and precipitation, the surface layer of proglacial lake is the first
to be affected, with enhanced hydrodynamic activity(Bagshaw et al., 2006; Li et al., 2022). Surface
eukaryotic microorganisms in the surface water diffuse under the environmental conditions shaped
by hydrodynamic forces(Stockwell et al., 2020; Zhu et al., 2023). As lake depth increases,
hydrodynamic activity diminishes, leading to a corresponding reduction in microbial diffusion
capacity(Burpee et al., 2018; Khan and Zutshi, 1980; Mohanty and Maiti, 2020). Therefore, the
diffusion ability of eukaryotic microorganisms in the surface layer of the lake was stronger, while
their biological drift ability was weaker, resulting in a lower biological drift ratio compared to the
bottom layer.

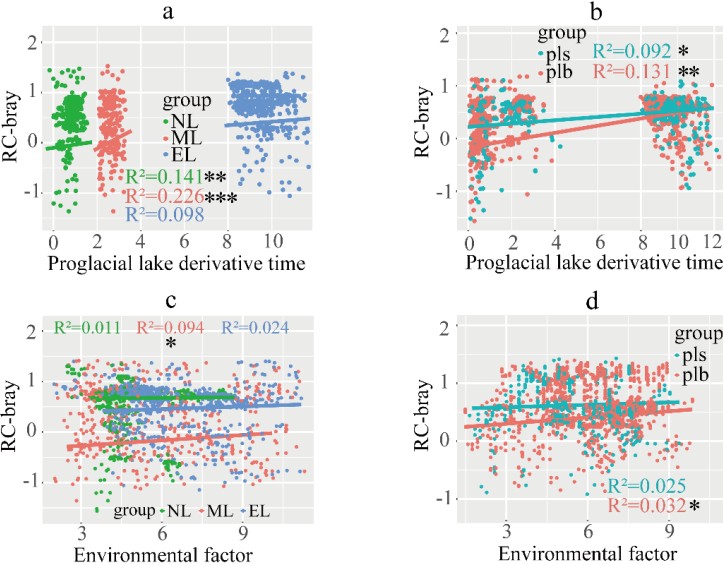


Fig. 8. Panels a and b illustrate the relationship between RC-ray and evolution time of proglacial
lake. Panels c and d depict the relationship between RC-ray and environmental factors. ***, p <
0.001; **p < 0.01; *p < 0.05.



(NL: Newly-emerging Lake; ML: Middle-period Lake; EL: Early-emerging Lake; pls: proglacial
lake surface; plb: proglacial lake bottom)

In general, stochastic processes highlight the influence of diffusion or ecological drift on

various in microbial community structure across different spatial and temporal scales(Dumbrell et
al., 2010). It is well established that when environmental selection is weak, communities are more
susceptible to ecological drift(Hubbell, 2001). Additionally, different components of stochastic
processes exert varying effects on eukaryotic microbial communities(Leibold and Mcpeek, 2006;
Stegen et al., 2015; Zorz et al., 2019). Among the three proglacial lakes, the proportion of
homogeneous diffusion is relatively low (Fig. 6). This may be attributed to the larger cell size of
eukaryotic microorganisms, which limits their diffusion capacity(Yang et al., 2023a). Among the
proglacial lakes, diffusion limitation is most pronounced in NL, followed by EL. Since NL is the
most recently developed proglacial lake with the shortest development period and direct connection
to glaciers, it exhibits the lowest water temperature and the longest freezing period throughout the
year, resulting in the most severe diffusion limitation. In contrast, EL is the oldest and most
developed proglacial lake, lacking any glacial supply or interference. Due to its low water exchange
rate, diffusion in EL is significantly restricted(Cauvy-Fraunié and Dangles, 2019; Cunde et al., 2006;
Wang et al., 2019). In summary, our study reveals the dynamics and mechanisms of water
environment changes during the evolution of proglacial lakes and the assembly process of
eukaryotic microbial community structure.
**5. Conclusion and Perspective**

This study utilized 18S rDNA gene amplicon sequencing and multiple statistical methods to

analyze the dynamic changes in eukaryotic microorganisms across various developmental stages of





proglacial lakes and KQGR at different levels. The diversity and structural composition of
eukaryotic microbial communities show clear spatial heterogeneity. As altitude decreases, microbial
diversity increases. Distance decay analysis indicates that differences in eukaryotic microbial
communities are strongly correlated with geographical distance. As the development time of
proglacial lakes increases and accelerated glacier retreat, the proportion of stochastic processes first
rises and then declines (ML > EL > NL). Proglacial lakes are located in climate-sensitive areas,
where glacier retreat due to warming, Proglacial lakes were connected to glaciers and were more
affected by glacial meltwater. This led to a higher proportion of stochastic processes (especially
ecological drift) in eukaryotic microorganisms in these lakes. As proglacial lakes developed and
glaciers retreated, they became increasingly separated. Consequently, proglacial lakes were
influenced by multiple factors, including glacial meltwater and runoff, while eukaryotic
microorganisms were increasingly shaped by stochastic processes. When the glacier had retreated
completely, the proglacial lake was less affected by factors such as surface runoff, causing
eukaryotic microorganisms to be less influenced by random processes. Therefore, during the initial
development stages of glacial lakes, glacial influence is most pronounced, and stochastic processes
dominate community assembly. As glaciers continue to retreat, the influence of stochastic processes
intensifies. However, once the glaciers have fully retreated, the influence of stochastic processes
diminishes. This suggests that the development of proglacial lakes and the evolution of glacial
retreat control stochastic processes that shape eukaryotic microbes.

This research broadens our understanding of the formation and ecological mechanisms

governing eukaryotic microbial communities in the aquatic systems of high-altitude proglacial lakes.
Analyzing the variability in co-occurrence networks and environmental factors offers valuable



insights into the mechanisms maintaining eukaryotic microbial communities. In high-altitude
ecosystems, proglacial lakes act as sentinels and recorders of climate change, showing heightened
sensitivity to shifts in environmental pressure. This makes them ideal environments for studying the
impact of climate change on ecosystems in high-altitude regions. Further research spanning a
broader geographical range and a longer historical timeframe of proglacial lakes is needed to better
understand the aquatic ecosystem's response to glacier retreat.
*Author contributions.* Y. Tong, Q. Zhang and J. Cui conceived the study. J. Cui, F. Mai, S. Li, M. Li,
J. Wang, X. Sun, Q. Zhang collected samples from the glacial lake. J. Cui and S. Li analyzed the
environmental factors. Q. Yang provided data analysis methods. J. Cui analyzed the data and
prepared the figures and tables. J. Cui, Y. Tong and Q. Zhang wrote the manuscript. All authors
edited and approved the final manuscript.
*Competing interests.* The contact author has declared that none of the authors has any competing
interests.
*Acknowledgements.* This work was supported by the second Tibetan Plateau Scientific Expedition
and Research Program (STEP), grant no. 2019QZKK0605, the National Natural Science Foundation
of China (42471161, 42276243, and 41922046). Q.Z. acknowledges financial support from the
Youth Innovation Promotion of CAS (202022).
*DATA AVAILABILITY.*  The data used in this study contain sensitive information and cannot be
shared publicly due to privacy concerns. But are available from the corresponding author on
reasonable request.



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
