# Peer review of "Distribution Patterns and Community Assembly Processes of Eukaryotic Microorganisms in"

_EGUsphere, 2024_

## Referee Comment (RC1)

This is a potentially interesting paper on microbial succession in those lakes that form at the base of retreating glaciers on the Tibetan Plateau. It is interesting, but not surprising that community structure could change over time in these lakes, but the over-selling of the resulst based on three lakes is a substantial weakness.

In general it is over-interpreted and provides speculations as results or even conclusions. It is too long, but does not explain concepts necessary for understanding what or why the authors are performing their analysis, but often over explains the well-known.

The English in places difficult to parse and there is far too much private use of acronyms making it difficult to read.

While the sample size of one lake per age group is not statistically useful, the study, if substantially shortened, would make a useful note.

Given the small sample size, it seems difficult to satisfy the study's listed aims (Lines 190-193).

Some line by line comments (incomplete as not everything problematic is detailed here):

Line 47 "B"

49 " "

50 two "founds"

51 add comma. round up 1185.9

53 mean? or median area?

54 define "proglacial"

55 about "700,000"

56 "or 7%"

61 18% over 1977-2001 or when?

Lines 48-64 numbers are distracting (but valuable!). Put in parentheses and use 1/3 ¼ etc.

Line 65 new paragraph

Line 108 what is "they" referring to?

Line 129-130 The authors do not provide enough review of the previous studies, Freimann et al., 2015 and Peter and Sommaruga to distinguish their study form these two previous ones. Much of what is written here in the introduction is too general. The earlier parts of the Intro can be trimmed to provide better context about the known biology and ecology of proglacial lakes.

Line 139 "In this context" repeated

Line 142 new paragraph

Line 150 what is "AM." Define acronyms at first use and if only used once then use words, not acronym.

Line 166+ seem redundant with previous introductory material.

Line 170 new paragraph

Lines 170-193 are good.

Line 199: use decimal degrees: latitude 29o5', longitude 90o12',

Line 202: what is annual precipitation? (i.e. how much snow + rainfall?)

Line 209: suggest making Fig S1 as main text Figure 1 instead.

On Fig S1, suggest putting names of glacier (Kuoqionggangri) and mountain (Mount Nyenchen Tanglha) on Fig S1 b.

Line 210 use decimal degrees: (latitude: 29°52'2"; longitude: 90°11'48")

Line 215 "Early" and "New " are similar. Why not "2010s lake", "1990s lake" and "pre-1980 lake"? or something more descriptive and simpler for the reader? The acronyms do not help. If the authors need to save on word count, they can easily reduce the length of the text by more concise writing and less redundancy.

Line 216 replace "significant" with "substantial" as there are only sample sizes of n = 1 for each lake type. This is the primary weakness with this study. It is a case study. The word "significant" should be reserved for statistical analysis.

Lines 222-223 is false in the cases of phosphorous, iron, and other trace micro-nutrient metals.

Lines 226+ Box plots show multiple samples, with no indication of sample size as far as I can find anywhere in the text.

Lines 231-232 acronyms that replace useful words do not enhance the readability and value of the study, but rather detract from it. "surface samples" and "bottom samples" are much better terms for communicating than pls and plb.

Lines 236: makes and models of the meters are usually reported along with addresses or websites.

Lines 239-241 should use standard notation. These acronyms are confusing.

Line 269: PCoA is non standard. Please use PCA.

Lines 270-271: these non-standard acronyms are not helpful. Please use the names of the tests.

Line 288: Explain how "phylogenetic turnover between samples" is calculated and what is it measuring?

Line 282: briefly, describe "framework of Stegen et al"

Line 309-313: this requires conceptual explanation. Please remove fluff and well-known material from introduction so that these specialized concepts can be explained simply and usefully. It will make for a much better paper.

**Results:** difficult to follow because authors have chosen to use private acronyms instead of standardized terms for common variables.

Poor or no explanations provided to help a reader understand the graphics.

Lines: 343-344 please inform readers by also including higher taxonomic names with what we readers can only infer as genera, since they are italicized. If they are genera, please provide, family, order, phylum, and kingdom at first mention in parentheses. If they are not genera, remove italics.

Line 344: The use of the word "species" here appears to this reviewer to be mistaken. For example, do not italicize Nematoda, if you are referring to the nematodes as a phylum, perhaps say something like Nematoda (Animalia)…or whatever is the current practice.

Line 352 is not a sentence. Needs a verb.

Move Fig 2 to supplement if there are no differences. It would also be very helpful to guide readers through these figures.

Line 365: consistent or similar results? In what way were they similar (or consistent)? IN a few words introduce what you are going to tell the reader.

Line 374: maybe say between lake variability was greater than within lake variability?

Fig 3c and d: what is "between"?

Line 387 Tell us in methods what Variance Partitioning Analysis (VPA) is, please.

Line 437: It appears that there are only 2 "different layers": perhaps simply say "showed negligible differences between surface and bottom layers"?

Lines 438-440 are confusing and unclear but seem somehow important.

Lines 442-443:  But they may not "result in". They may be correlated with but correlation is not causation, and the words "result in" suggest causation.

Lines 445-447: Because the concepts of these analyses and graphics were not explained, this reviewer must disagree: what shows "synergistic symbiosis"? If I always see many wolves with many reindeer, does that mean there is a synergistic symbiosis?

Lines 419+ This sort of analysis would be believable if multiple samples of each lake age were used. Otherwise this analysis is only comparing three nearby lakes that may differ for other reasons besides age per se. It's impossible to distinguish spatial vs temporal processes in this study given its experimental design.